# Engineering surface atomic structure of single-crystal cobalt (II) oxide nanorods for superior electrocatalysis

Tao Ling[1,2,*], Dong-Yang Yan[1,*], Yan Jiao[2,*], Hui Wang[3], Yao Zheng[2], Xueli Zheng[1], Jing Mao[1], Xi-Wen Du[1], Zhenpeng Hu[4], Mietek Jaroniec[5] & Shi-Zhang Qiao[1,2]

Engineering the surface structure at the atomic level can be used to precisely and effectively manipulate the reactivity and durability of catalysts. Here we report tuning of the atomic structure of one-dimensional single-crystal cobalt (II) oxide (CoO) nanorods by creating oxygen vacancies on pyramidal nanofacets. These CoO nanorods exhibit superior catalytic activity and durability towards oxygen reduction/evolution reactions. The combined experimental studies, microscopic and spectroscopic characterization, and density functional theory calculations reveal that the origins of the electrochemical activity of single-crystal CoO nanorods are in the oxygen vacancies that can be readily created on the oxygen-terminated {111} nanofacets, which favourably affect the electronic structure of CoO, assuring a rapid charge transfer and optimal adsorption energies for intermediates of oxygen reduction/evolution reactions. These results show that the surface atomic structure engineering is important for the fabrication of efficient and durable electrocatalysts.

[1] Tianjin Key Laboratory of Composite and Functional Materials, Key Laboratory for Advanced Ceramics and Machining Technology of Ministry of Education, Institute of New-Energy, School of Materials Science and Engineering, Tianjin University, Tianjin 300072, China. [2] School of Chemical Engineering, The University of Adelaide, Adelaide, South Australia 5005, Australia. [3] Key Laboratory of Aerospace Materials and Performance (Ministry of Education), School of Materials Science and Engineering, Beihang University, Beijing 100191, China. [4] School of Physics, Nankai University, Tianjin 300071, China. [5] Department of Chemistry and Biochemistry, Kent State University, Kent, Ohio 44242, USA. * These authors contributed equally to this work. Correspondence and requests for materials should be addressed to S.-Z.Q. (email: s.qiao@adelaide.edu.au) or to X.-W.D. (email: xwdu@tju.edu.cn) or to Z.H. (email: zphu@nankai.edu.cn).

The growing concerns over climate change and energy security have stimulated a rapid development in the generation of clean energy[1]. Electrocatalysts play a key role in sustainable energy production, including fuel cells[2], metal-air batteries[3–5] and water splitting[6]. Currently, noble metals and their complexes are the most efficient catalysts[7–9], but their high cost and scarcity greatly restricts commercial applications[10]. Transition metal oxides (TMOs) are considered as the most probable alternatives to noble metal-based catalysts due to their low cost, nontoxicity and high stability[3,11–19]. Nevertheless, there is still an urgent need to further improve their activities and make them highly competitive in comparison with their noble-metal counterparts and useful for practical applications.

In the light of the above discussion, the surface structure engineering of TMO catalysts involving better exposure of active sites to promote their electrocatalytic performance becomes of paramount significance. In the past decade, fundamental research has demonstrated that the rational design of active facets with favourable atomic arrangement and coordination is the most promising route to control the atomic structure of noble metals[7,8] and metal oxide[20–24] particulate catalysts, and achieve high catalytic activity. Further investigations suggest that the surface defects[25–28] can greatly influence the electronic structure and thus the surface chemistry of faceted catalysts[29–36]. Hence, a proper manipulation of defects on the desired facets of catalysts has received a considerable attention and brought some exciting breakthroughs. For instance, introduction of defects on {100} facets has been effectively used to tailor the band structure of titanium dioxide ($TiO_2$) to make it suitable for photocatalytic hydrogen evolution under visible light illumination[30,33]. Engineering sulfur vacancies on the basal planes of molybdenum disulfide ($MoS_2$) nanosheets can be used to finely tune the adsorption free energy of hydrogen to achieve the highest activity of the aforementioned nanosheets towards hydrogen evolution reaction among various $MoS_2$-based catalysts[31]. However, the application of faceted TMO catalysts in electrocatalysis is in its infancy[37]. Also, engineering favourable defects on the desired facets and understanding their role in electrocatalysis at the atomic level is still lacking.

Very recently, one-dimensional (1D) nanoarrays directly grown on the current collectors have attracted a lot of attention in electrocatalysis[16,38–42] because their 1D morphology assures adequate diffusion of reactants and rapid charge transport. Although a great progress has been achieved in electrocatalysis, much less has been done towards engineering the surface atomic structure of the aforementioned nanoarrays to explore their full potential. Herein, we report the surface structure engineering of single-crystal (SC) CoO nanorods (NRs) through creating desired facets and defects (Fig. 1). Our experiments, microscopic and spectroscopic characterization and the density functional theory (DFT) computation studies demonstrate that the O-vacancies present on the pyramidal nanofacets of CoO NRs can be effectively used to tailor the electronic structure of NRs, which results in rapid charge transfer and favourable energetics for both oxygen reduction reaction (ORR) and oxygen evolution reaction (OER) as evidenced by excellent activity and durability of CoO NRs towards both reactions. Significantly, their ORR activity approaches that of platinum (Pt) catalysts and their OER activity exceeds that of ruthenium dioxide ($RuO_2$) catalysts; their overall activity is comparable to that of the best bifunctional ORR/OER catalysts.

## Results

### Synthesis and characterization of SC CoO NRs. First, we report the synthesis of SC CoO NRs with textured pyramidal nanofacets

immobilized directly on a carbon fibre paper (CFP) substrate (Fig. 1a–c) via a simple and well-controlled cation exchange method (Supplementary Fig. 1). This synthesis was accomplished by converting SC zinc oxide (ZnO) NRs (Supplementary Fig. 2) to CoO NRs (Supplementary Fig. 3) via the aforementioned cation exchange reaction in gas phase[43]. A controlled fabrication of CoO NRs with tailorable length was achieved by precise tuning the length of ZnO NRs from dozen nanometres to several microns (Supplementary Fig. 4).

The microstructure of as-synthesized CoO NRs was investigated by scanning electron microscopy (SEM) and transmission electron microscopy (TEM). As displayed in Fig. 2a, the entire surface of CFP is uniformly covered with CoO NRs, which are SCs (Fig. 2c, inset). Interestingly, after cation exchange reaction, numerous nanopores with sizes of 5–20 nm are visible on the surface and across NRs (Supplementary Fig. 5). Surprisingly, the surface of CoO NRs becomes rather rough as evidenced by tooth-like growths with sizes of about 5 nm (Fig. 2b). Atomic level high-angle annular dark field-scanning TEM (HADDF-STEM) image shows that these growths are sharply terminated with {111} nanofacets (Fig. 2c). Notably, the gradual contrast variation in these single tooth-like growths suggests a progressive variation in their thickness (Fig. 2c). Simulation of experimental image was performed to accurately determine the three-dimensional atomic arrangement in the aforementioned growths. A speculated nanopyramidal structure with exposed {100} and {111} facets was constructed as shown in Fig. 2d,e. Figure 2f,g indicates a good agreement between the experimental and simulated images. Moreover, the intensity profile along the terminated {111} facet in the experimental image (Fig. 2h) closely resembles that in the simulated one (Fig. 2i). A good match between experimental and simulated HADDF-STEM images clearly demonstrates that the surface of SC CoO NRs is surrounded by nanopyramids and preferentially exposed {111} facets. The surface area of exposed {111} facets is estimated to be 46% of the total surface area of SC CoO NRs (Supplementary Fig. 6 and Supplementary Note 1). Notably, for CoO the surface energy of {111} is much higher than that of other low-indexed facets[44]. Such high percentage of {111} facets without foreign stabilizer is rather difficult to achieve via thermodynamically controlled synthesis[44]. However, in our kinetics-governed cation exchange strategy, facets with high surface energy and defects (discussed later) are forced to be exposed to facilitate the ion exchange process[45], assuring the formation of a large amount of clean and defect-rich {111} facets on the surface of SC CoO NRs, which is certainly highly preferable for catalysis. Furthermore, it should be noted that {111} facets of the bulk CoO are polar, either terminated by oxygen (O) or Co-atomic layer[46]. A detailed X-ray photoelectron spectroscopy (XPS) analysis indicates that the exposed {111} facets on CoO NRs should be O-terminated (Supplementary Fig. 7, Supplementary Table 1 and Supplementary Note 2).

### Analysis of O-vacancy-rich pyramidal nanofacets. To further probe the local chemical and electronic environment on the surface of SC CoO NRs, XPS and synchrotron-based X-ray absorption near edge fine structure (XANES) spectroscopy measurements were performed. The XPS O 1s spectrum suggests an enrichment of O-vacancies on the surface of SC CoO NRs (Supplementary Fig. 7a). Further evidence comes from a close inspection of the fine structure of the O-K edge of XANES spectrum (Fig. 3a), in which the peak at ~536.0 eV assigned to O deficiency[47,48] in SC CoO NRs is much higher than that of reference CoO. This is also consistent with the observation of a noticeable peak shift in Co-$L_{2,3}$ edge towards low photon energy and Co 2p XPS spectrum towards low binding energy of SC CoO

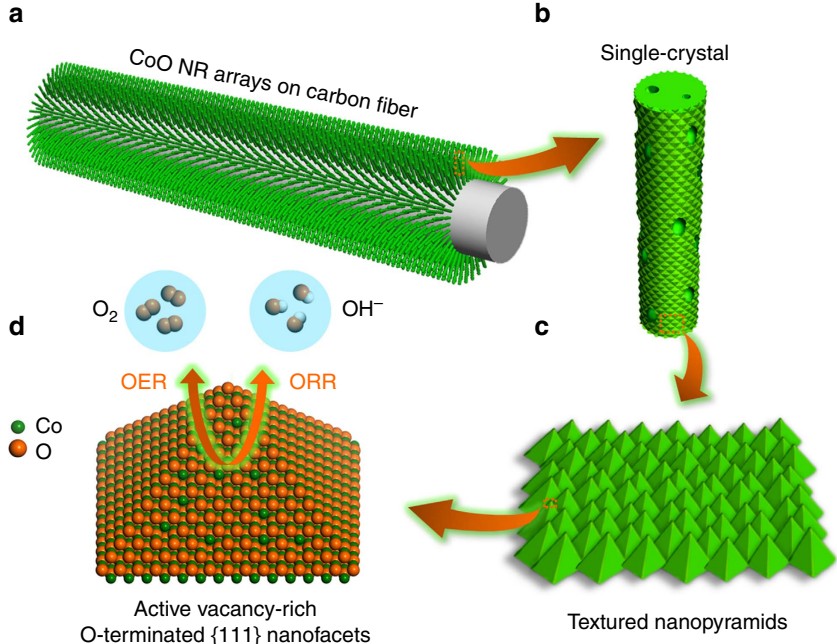

**Figure 1 | Schematic illustration of engineering the surface of SC CoO NRs.** (**a**) SC CoO NRs fabricated directly on carbon fibre substrate. (**b**) Numerous nanopores present on the surface and across SC NRs. (**c**) The surface of SC CoO NRs covered with textured nanopyramids. (**d**) The dominant exposed facets of nanopyramids are electrochemically active vacancy-rich O-terminated {111} facets.

NRs (Fig. 3b and Supplementary Fig. 8), which is an indicative of electron transfer from O-vacancies to Co $d$ band. Thus, a combination of XANES and XPS results provides a crucial evidence for the presence of abundant O-vacancies on the surface of SC CoO NRs, the quantity of which is even higher than that on the surface of polycrystalline (PC) CoO NRs with threefold larger surface area (Supplementary Table 1). Our DFT computations indeed reveal that the O-vacancy formation energy on the O-terminated {111} facets (hereafter, referred to as '{111}-O facet') is by 3 eV lower than the corresponding values of {100} and {110} facets (Fig. 3c). Clearly, such a significant reduction in the vacancy formation energy results in larger concentration of the equilibrium O-vacancies on {111}-O facets of SC CoO NRs. It indicates that the surface defects can be tuned and stabilized through facet engineering.

**Activity and durability of SC CoO NRs towards ORR/OER.** As-fabricated SC CoO NRs with length of 1.6 µm on CFP were directly used as the working electrodes for both ORR and OER (Supplementary Fig. 9), and their performances were compared with analogous electrodes prepared by using PC CoO NRs (Supplementary Fig. 10), the state-of-art Pt and RuO$_2$ catalysts supported on CFP. The polarization curves were recorded without *iR* correction. As regards ORR (Fig. 4a,b and Supplementary Fig. 11a), PC CoO NRs exhibit low activity, while SC CoO NRs show an onset potential of 0.96 V versus reversible hydrogen electrode (V$_{RHE}$), a half-wave potential ($E_{1/2}$) of 0.85 V$_{RHE}$ and a Tafel slope of 47 mV per decade, which approach the values measured for Pt catalysts. These values are better than those of the well-developed cobalt oxide nanocrystals (NCs) coupled with carbon materials (Supplementary Table 2). Moreover, SC CoO NRs show high selectivity towards ORR with strong methanol tolerance (Fig. 4a, inset). Besides an extraordinary activity towards ORR, SC CoO NRs also demonstrate an excellent durability. As shown in Fig. 4c, SC CoO NRs retain 97% of the initial ORR current after 10 h continuous testing, whereas Pt catalyst lost more than 26% of its initial current, confirming much

better durability of active reaction sites present on SC CoO NRs (Supplementary Figs 12 and Supplementary Fig. 13). Moreover, even after 3,000 cycle catalytic tests with accelerated scan rate of 100 mV s$^{-1}$, SC CoO NRs still retained their structure (Supplementary Fig. 14). This excellent durability originates from direct growth of SC CoO NRs on the CFP substrate to avoid aggregating and detaching problems, which are usually encountered in other faceted catalysts[49].

As regards the OER activity (Fig. 4d), SC CoO NRs deliver a current density of 10.0 mA cm$^{-2}$ ($E_{J=10}$) at 1.56 V$_{RHE}$ and a Tafel slope of 44 mV per decade. Such excellent OER activity is better than that of the commercial RuO$_2$ catalyst. Importantly, SC CoO NRs exhibit an outstanding overall electrode activity as indicated by lower value ($\Delta E = 0.71$ V) of the difference between the ORR and OER metrics ($\Delta E = E_{J=10} - E_{1/2}$)[16,50], outperforming the most of the reported highly active reversible oxygen catalysts (Supplementary Table 2). This value is also comparable to the value obtained for Co$_3$O$_4$ NCs deposited on N-doped graphene ($\Delta E = 0.71$ V)[11], which is considered as the most efficient bifunctional catalyst.

**Enhancement of electronic conductivity of SC CoO NRs.** High activity and durability of SC CoO NRs clearly demonstrates that the as-synthesized SC CoO NRs are highly versatile and efficient electrocatalysts towards both ORR and OER. One prerequisite in the design of a highly efficient electrocatalyst is a rapid electron transfer[16]. It is acknowledged that the electronic conductivity of TMOs is relatively poor[15], greatly limiting their electrocatalytic activities. In the case of SC CoO NRs, a large quantity of O-vacancies localized on the {111}-O facets, as well as the SC nature inherited from the ZnO NRs considerably enhance the carrier concentration in SC CoO NRs, which is one order higher than that in PC CoO NRs (Supplementary Fig. 15 and Supplementary Note 3). Moreover, the nucleation and growth of SC CoO NRs directly on CFP also assures a rapid collection of charges. The aforementioned three important characteristics remarkably enhance the electronic conductivity of SC CoO NRs.

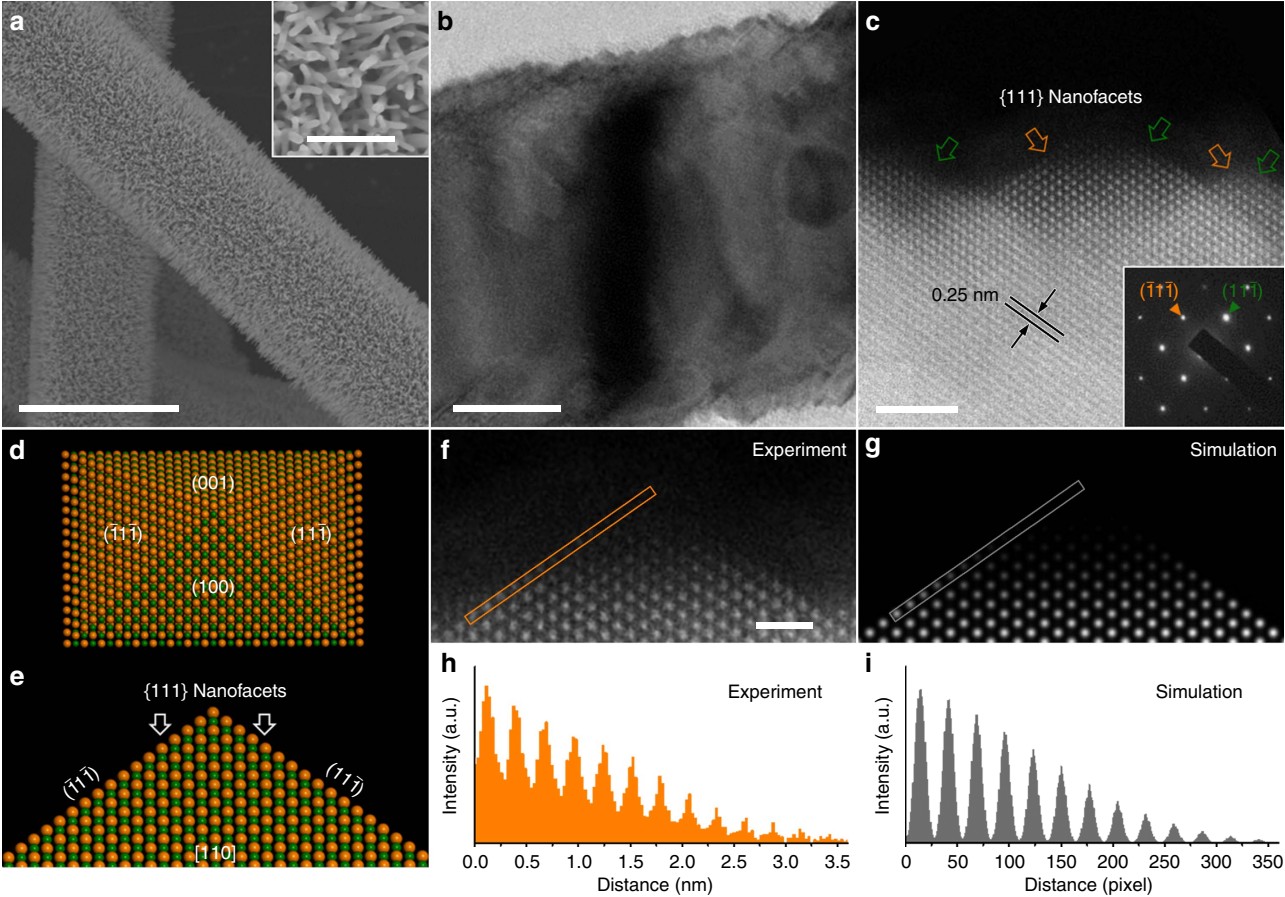

**Figure 2 | Structural characterization of SC CoO NRs.** (**a**) Top-view SEM image of SC CoO NRs fabricated directly on CFP. Scale bar, 10 µm. The inset in **a** shows morphology of SC CoO NRs. Scale bar, 1 µm. (**b**) High magnification TEM image of an individual SC CoO NR with saw-like edges. Scale bar, 20 nm. (**c**) High-resolution HAADF-STEM image taken from the outermost surface of a single SC CoO NR revealing the exposed {111} nanofacets (indicated by orange and green arrows), with inset showing the corresponding selected area electron diffraction pattern taken from [110] zone axis. Scale bar, 2 nm. (**d**) Atomic model of a nanopyramid enclosed with {100} and {111} facets, and (**e**) the projection of this pyramidal structure along [110] zone axis. (**f,g**) Experimental and simulated HADDF-STEM images of the pyramidal structure, respectively. Scale bar in **f**, 1 nm. Note that inelastic and neutron scatterings were not considered in the simulation, which contribute to the background in **h**. (**h,i**) The intensity profiles taken from orange and grey lines in **f** and **g**, respectively.

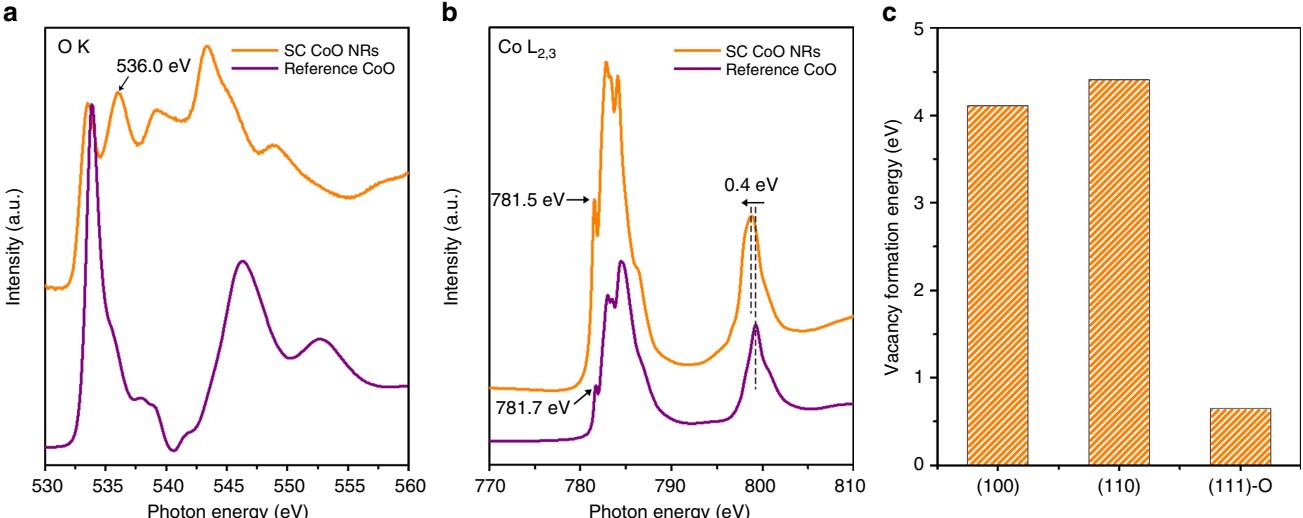

**Figure 3 | Analysis of O-vacancies on the pyramidal nanofaceted surface of SC CoO NRs.** (**a,b**) O-K edge and Co-L$_{2,3}$ edge XANES spectra of SC CoO NRs and reference CoO, respectively. In **a** the peak at ∼536 eV of SC CoO NRs is assigned to O deficiency. In **b** the peaks at 781.5 and ∼800 eV of SC CoO NRs shift towards low photon energy relative to the reference CoO, indicating the transfer of electrons from O-vacancies to Co $d$ band. (**c**) O-vacancy formation energies on {100}, {110} and {111}-O facets of CoO showing a significant reduction in the vacancy formation energy on {111}-O facets.

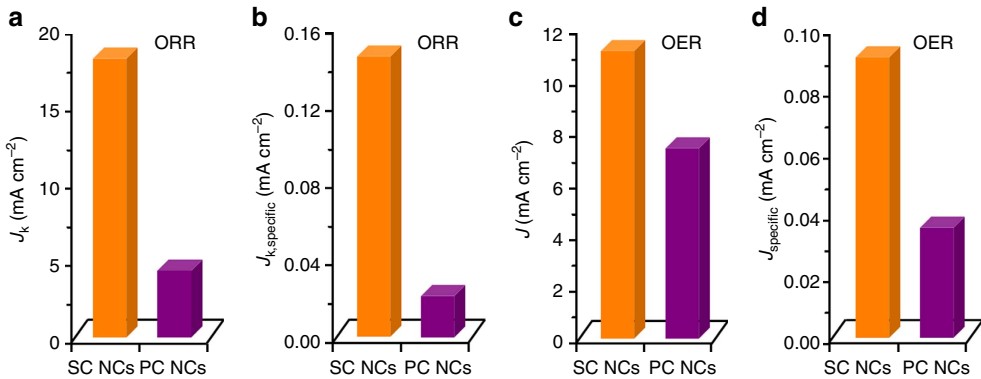

**Figure 4 | Bifunctional ORR/OER performance of SC CoO NRs.** (**a**) ORR linear-sweep voltammograms (LSVs) of SC, PC CoO NRs and Pt catalysts directly deposited on CFP in $O_2$-saturated 1 M KOH solution at scan rate of $0.5\,mV\,s^{-1}$ without *iR* correction, with ORR chronoamperometric response to methanol addition shown in inset. (**b**) ORR Tafel plots of SC, PC CoO NRs and Pt catalysts. (**c**) ORR chronoamperometric response of SC CoO NRs and Pt catalysts at a constant voltage of $0.60\,V_{RHE}$. (**d**) OER LSVs of SC, PC CoO NRs and commercial $RuO_2$ catalysts directly deposited on CFP in $O_2$-saturated 1 M KOH solution at scan rate of $0.5\,mV\,s^{-1}$ without *iR* correction, with the corresponding OER Tafel plots shown in inset.

**Figure 5 | Intrinsic ORR/OER activity of SC CoO NCs in comparison with the activity of PC CoO NCs.** (**a,b**) $J_k$ and $J_{k,specific}$ for ORR at $0.6\,V_{RHE}$, respectively. (**c,d**) $J$ and $J_{specific}$ for OER at $1.65\,V_{RHE}$, respectively.

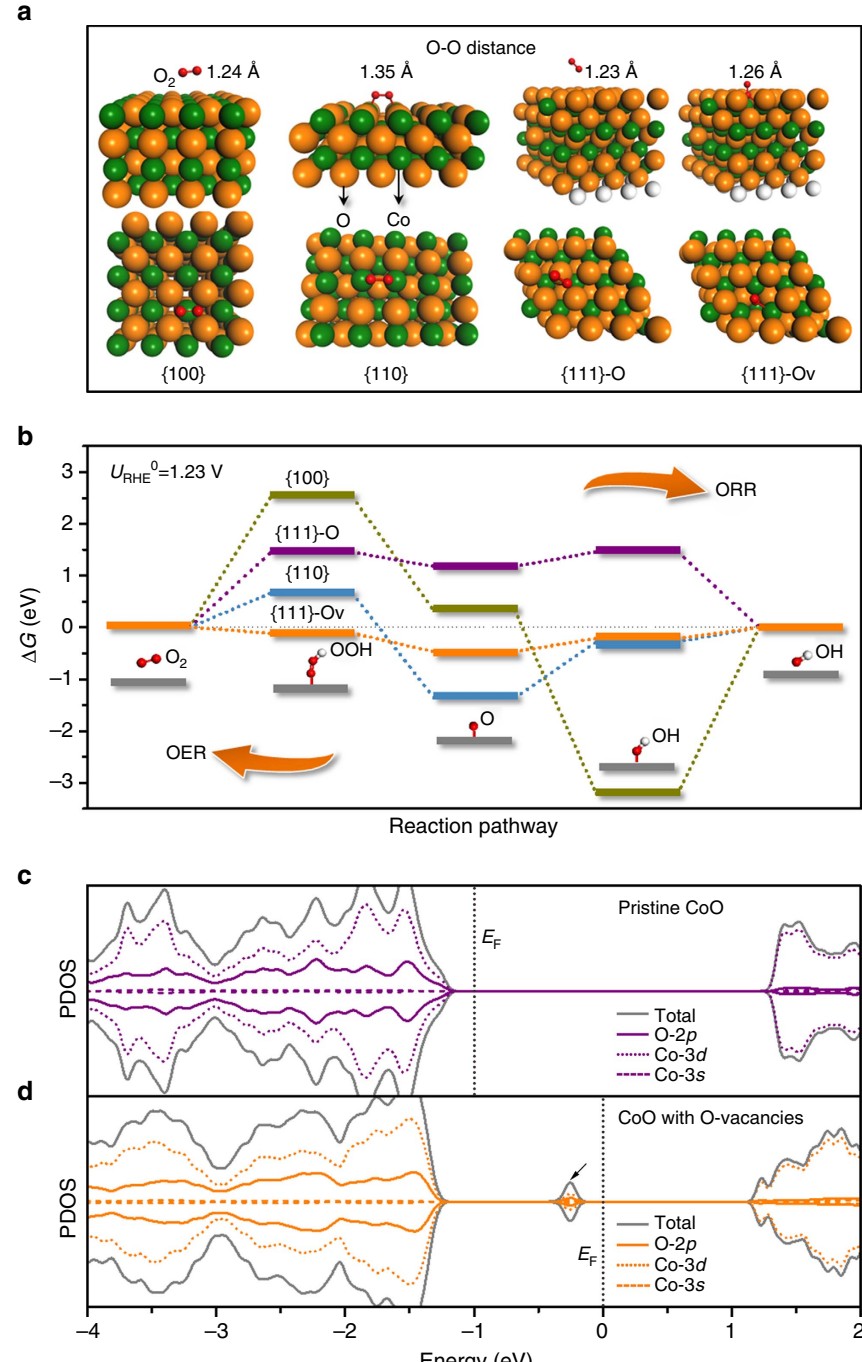

**Figure 6 | Origin of ORR/OER activity on various facets of CoO.** (**a**) Atomic configurations of $O_2$ molecules on {100}, {110}, {111}-O and {111}-$O_V$ facets. Notably, on {110} and {111}-Ov facets, the O–O bond of adsorbed $O_2$ is remarkably elongated (the O–O distance of $O_2$ is 1.23 Å), suggesting an effective activation of $O_2$ in ORR. (**b**) The calculated ORR/OER free energy diagram at the equilibrium potential on different facets. (**c**) The projected density of states (PDOS) on pristine CoO and (**d**) CoO with O-vacancies. The arrow in **d** points new electronic states, which appear near the Fermi level in CoO with O-vacancies, responsible for the adsorption of intermediates on the O-vacancies.

**Intrinsic activity of SC CoO NCs towards ORR/OER.** To decouple the enhanced activity of SC CoO NRs from the contribution of advanced 1D nanoarray architecture, the intrinsic ORR/OER activity of SC CoO NCs with sizes of $\sim 50$ nm (Supplementary Method) was evaluated in comparison to that of PC CoO NCs (Supplementary Figs 16–18 and Supplementary Note 4) and other well-developed particulate cobalt oxide catalysts (Supplementary Table 6). The ORR kinetic current and OER current are normalized by the electrochemically active surface area of catalyst to obtain the specific ORR kinetic current density ($J_{k,specific}$) and the specific OER current density ($J_{specific}$), respectively (see definition in Supplementary Table 3). As shown in Fig. 5a,b, for ORR, $J_k$ and $J_{k,specific}$ of SC CoO NCs at 0.6 $V_{RHE}$ are about 4.2 and 7.2 times greater than those of PC CoO NCs, respectively. As regards OER, $J$ and $J_{specific}$ of SC CoO NCs at 1.65 $V_{RHE}$ are about 1.5 and 2.6 times larger than those of PC CoO NCs, respectively (Fig. 5c,d). These collective results clearly demonstrate that the intrinsic ORR/OER activity of CoO is strongly dependent on the surface structure.

**ORR/OER free energy diagram and electronic structure.** A series of DFT computations was conducted to get a fundamental understanding of the correlation between the surface atomic structure of CoO and the ORR/OER activity. Fig. 6a clearly reveals that the atomic arrangement on different facets significantly affects adsorption sites and configuration of the reactant, that is, $O_2$, in ORR. Moreover, the overall ORR/OER pathway was calculated, and the free energy diagram at the equilibrium potential ($U_{RHE}^0 = 1.23$ V) are shown in Fig. 6b. As reported by Nørskov et al.[51–53], both ORR and OER involve four elementary reaction steps, in which ORR proceeds through the formation of HOO$^\star$ from adsorbed $O_2$, followed by its further reduction to O$^\star$ and HO$^\star$, while OER proceeds in the reverse direction. For both ORR and OER, the ideal thermodynamic free energy change of the intermediates should be $|\Delta G_{OOH*}| = |\Delta G_{O*}| = |\Delta G_{OH*}| = 0$ (refs 51,52), indicating no energy would be wasted to activate the reactions. As illustrated in Fig. 6b, for ORR a large $|\Delta G_{OOH*}|$ on the surface of {100} and {111}-O facets indicates that the first electron transfer step to reduce the adsorbed $O_2$ to OOH$^\star$ is endothermic, which is consistent with observations for the well-developed metal and metal oxide catalysts[17,53]. Besides, the large negative $\Delta G_{O*}$ and $\Delta G_{OH*}$ on {110} and {100} facets indicate that the chemical adsorption of O$^\star$ and OH$^\star$, respectively, is too strong, which is also unfavourable for the subsequent electrocatalytic reactions. However, when O-vacancy is created on the surface of {111}-O (hereafter, referred to as '{111}-$O_V$ facet'), the formation of OOH$^\star$ is facilitated, and all $|\Delta G_{OOH*}|$, $|\Delta G_{O*}|$ and $|\Delta G_{OH*}|$ exhibit the lowest values among the four facets, suggesting the most favourable ORR kinetics on the {111}-$O_V$ facets. As regards OER, a similar analysis of the diagram for the reverse reaction shows that the {111}-Ov surface outperforms the other three facets. Overall, the {111}-Ov surface exhibits a mediated adsorption–desorption behaviour ($|\Delta G_{OOH*}| \approx |\Delta G_{O*}| \approx |\Delta G_{OH*}| \rightarrow 0$), which is beneficial for the overall ORR/OER. Thus, the theory and experiment are in an excellent agreement, suggesting that the ORR/OER activity is successfully enhanced through atomic structure engineering. Our results demonstrate that there is a strong correlation between activity and atomic structure of CoO; that is, the ORR/OER activity of CoO increases in the following order {100} < {110} < {111}-Ov. To our knowledge this atomic scale structure–function relationship has not been considered for any other TMO surfaces in the analysis of electrocatalysts.

The nature of the O-vacancies on the {111}-O facets is further revealed through investigation of their electronic structure (Fig. 6c,d). As indicated by the arrow in Fig. 6d, when O-vacancy is created in CoO, some new electronic states are created by hybridization of O-2$p$, Co-3$d$ and Co-3$s$ in the bandgap, which are directly responsible for stronger adsorption of intermediates on the O-vacancies and for higher electronic conductivity of CoO (Supplementary Fig. 19). Therefore, optimization of the electrocatalytic activity of CoO through creation O-vacancies on the {111}-O facets can be finally ascribed to the successful engineering of the electronic structure of CoO.

## Discussion

In conclusion, SC CoO NRs with exposed vacancy-rich pyramidal nanofacets were successfully fabricated on the CFP support. The current experimental and theoretical study shows that the creation of O-vacancies on the {111}-O facets favourably affects the electronic structure of CoO, resulting in the enhanced charge transfer and optimal energetics for both ORR and OER, which boosts the overall electrode activity of SC CoO NRs. Our work indicates that the rational design of the atomic surface of

nanoarray electrodes can pave a new avenue for the fabrication of efficient and durable TMO-based electrochemical devices.

## Methods

**Synthesis of SC CoO NRs on CFP substrate.** ZnO NRs with tailorable length (Supplementary Figs 4 and 20) were grown on CFP under hydrothermal conditions and finally converted into CoO NRs using a cation exchange process in gas phase (Supplementary Fig. 1). Specifically, the CFP loaded with ZnO NRs was placed in the centre of a quartz tube and cobalt chloride ($CoCl_2$) powder was placed 2.5 cm upstream from the tube centre. After the quartz tube was outgassed under vacuum, argon (Ar) gas flow (50 s.c.c.m.) was introduced into the system. The furnace was heated to and kept at 600, 650 or 700 °C for 30 min, and then cooled down to room temperature. It is found that the cation exchange temperature can considerably affect the concentration of O-vacancies (Supplementary Table 7), and thus the electrocatalytic performance of SC CoO NRs (Supplementary Figs 21–23 and Supplementary Note 5). The optimal exchange temperature is 600 °C and used in this study unless specifically notified. The loading mass of as-synthesized SC CoO NRs was $\sim 0.19$ mg cm$^{-2}$.

**Characterization.** SEM and TEM images were taken on a Hitachi S-4800 SEM and a JOEL 2100 TEM, respectively. HAADF-STEM images were collected using a JEOL ARM200F microscope with STEM aberration corrector operated at 200 kV. HADDF-STEM image simulation was carried out using a software package MacTempasX. The convergent semiangle and collection angle were 21.5 and 200 mrad, respectively. The aberration coefficient ($C_s$) used was equal to 1 μm. Inelastic and neutron scatterings were not considered in the simulation. The synchrotron-based XANES measurements were carried out using the soft X-ray spectroscopy beamline at the Canadian Synchrotron. XANES spectra were recorded in the surface sensitive total electron yield with use of specimen current. All samples were scanned from 750 to 820 eV and from 510 to 580 eV in 0.1 eV steps, which encompasses the Co-L$_{2,3}$ and O-K absorption edges, respectively.

**Electrochemical characterization.** Electrochemical measurements were performed in a three-electrode electrochemical cell using an Hg/HgO electrode in saturated KCl solution as the reference electrode, Pt plate as the counter electrode and the CFP electrode as the working electrode (Supplementary Fig. 9). A flow of $O_2$ was maintained over the electrolyte (1.0 M KOH) during measurements to ensure the $O_2/H_2O$ equilibrium at 1.23 V$_{RHE}$.

**DFT calculations.** All DFT computations were performed using Vienna Ab-initio Simulation Package. An effective $U$ value of 3.7 eV was applied for Co 3$d$ states. The projector augmented wave pseudopotential with the Perdew–Burke–Ernzerhof exchange-correlation functional was used in the computations. The relevant details, references and data are given in the Supplementary Methods section and in Supplementary Tables 4 and 5.

**Data availability.** The data that support the findings of this study are available from the corresponding author on request.

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

## Acknowledgements

This work was supported by the National Basic Research Program of China (2014CB931703), the Natural Science Foundation of China (51571149, 51471115, 21576202 and 21203099), the Natural Science Foundation of Tianjin city (15JCYBJC18200 and 13JCYBJC36800) and the Australian Research Council (ARC) through the Discovery Project programme (DP130104459, DP140104062 and DP160104866). DFT calculations were undertaken on Special Program for Applied Research Super Computation of the NSFC-Guangdong Joint Fund (the second phase). We would like to thank Ning Lu for his kind help in this work.

## Author contributions

T.L., X.-W.D. and S.-Z.Q. conceived the project and designed the experiments; T.L. and D.-Y.Y. performed the experiments; T.L., H.W. and J.M. carried out the TEM characterization, HADDF-STEM image simulation and analysis; X.Z. performed the XANES characterization; Y.J., Z.H. and T.L. conducted the DFT calculations. All authors discussed the results and commented on the manuscript.

## Additional information

**Competing financial interests:** The authors declare no competing financial interests.

