## [Peer review file · Nature Communications]

Reviewers' Comments:

Reviewer #1 (Remarks to the Author)

In this manuscript, the authors report the surface atom structure tuning of single-crystal CoO nanorods through creating desired facets and oxygen defects, which show superior ORR and OER activities. Both experiments and calculations have been applied to explain the performances. This study is novel and very interesting. The data and methodology are reliable and the manuscript is also well organized and presented clearly. The conclusion is also clear and reliable. This manuscript can be considered for publication after addressing the following issues.

1. In Figure S9, the "PS CoO NRs" should be replaced by "SC CoO NRs"
2. Figure 4c shows that SC CoO NRs have a better durability than Pt catalyst. However, as shown in Figure S10, after several hours testing the surface state has changed slightly. I want to know whether much O-vacancies can still be maintained in oxygen-saturated KOH. The O1s spectra of SC CoO NRs after several hours should also be supplied to compare the surface state.
3. The samples prepared by using 3h ZnO and 6h ZnO have the same length, but why SC CoO NRs using 3h ZnO exhibit the lowest onset potential and largest OER current as shown in Figure S16.
4. In table S1, the atomic ratios of Co:O, Co:OC and Co:OL for SC and PC CoO NRs are quite different. The authors should add more discussions about them.
5. Some references about the effects of oxygen vacancy or crystal plane on electrocatalytic activity are missed. For examples: (1) *Angew. Chem., Int. Ed.* 2013, 52, 2474–2477. (2) *J. Mater. Chem. A* 2015, 3, 17598–17605. (3) *Nat. Chem.* 2012, 4, 1004–1010. (4) *ACS Catal.* 2016, 6, 400–406; (5) *Sci. Rep.* 2014, 4, 5767; (6) *J. Phys. Chem. C* 2015, 119, 4516.

Reviewer #2 (Remarks to the Author)

Review Report for "Engineering surface atomic structure of single crystal CoO nanorods for superior electrocatalysis", T. Ling, D.Y. Yan, Y. Jiao, H. Wang, Y. Zheng, X. Zheng, J. Mao, X.W. Du, Z. Hu, M. Jaroniec, and S.Z. Qiao

In the manuscript, the authors reported their experimental and computational results on electrocatalytic activity of single-crystal CoO nanorods terminated by O-vacancy-rich (111) facets for oxygen reduction reactions (ORR) and oxygen evolution reaction (OER) in alkaline solutions. Specifically, the authors had synthesized the CoO nanorods from ZnO nanorods using a cation exchange method, characterized the surface structure of the CoO nanorods using advanced electron microscopy and spectroscopy, and measured the catalytic activity of the fabricated CoO nanorods for ORR/OER using three electrode cell. Moreover, the authors performed density functional theory (DFT) calculations to predict the free energy evolution of ORR/OER on various CoO surfaces and the electronic structure of the CoO (111) surface with/without O vacancies. The authors concluded that the introduction of surface O vacancies could improve the electrocatalytic activity of CoO nanostructures. It is commendable that the authors have conducted this study in a meticulous way. However, the presented results only add some incremental knowledge/data to the current understanding of electrocatalysis. This reviewer does not believe the current manuscript contain enough innovative, significant contents to be considered for publication in *Nature Communications*.

My major criticisms to the manuscript are given in below.

1. It is unclear what the significances of this presented work are. (1) As revealed in Fig. 4, the electrocatalytic activities of the synthesized CoO nanorods are worse than the state-of-the-art electrocatalyst systems. Especially, the authors studied the electrocatalytic properties of the CoO

nanorods only in alkaline solutions, in which many other non-precious catalysts exhibit similar activities for ORR/OER. (2) The employed experimental and computational techniques are widely used in the catalyst study. (3) The authors could not precisely control the concentration and distribution of the O vacancies on the surface of the CoO nanorods. (4) CoO is a semiconductor material which is not the best choice for electrocatalysts that requires rapid electron transfer.

2. It is not a novel idea to tune the properties of semiconductor nanostructures through modification of their surface defects. Specifically for electrocatalysts, there are two main mechanisms for performance improvement. Surface defects might (a) enhance the electrical conductivity and (b) alter the paths of chemical reactions. The authors mentioned both effects in the manuscript but did not carry out sufficient study to quantify the two effects and pin down the main reason for the observed electrocatalytic activity enhancement. Therefore, this reviewer does not think the current manuscript is novel enough to be of interests to a broad audience of scientific community.

3. This manuscript does not contain durability test results for the CoO nanorods. It happens frequently that so-called "surface engineered" nanocatalysts exhibit initial good activity but lose their activity rapidly in electrochemical cycling tests.

4. Regarding the DFT computation, I have several questions about some deficiencies related to the computational details presented in the manuscript and Supplementary Information.

(a) It has been reported that spin-polarization is needed for surface adsorption calculation. The authors did not mention if they include this in their DFT calculations.

(b) It is unclear how the authors compared the predicted electrocatalytic activities for ORR/OER on different CoO surfaces from Fig. 6(b). At the given potential 1.23 V, all the electrochemical reactions should have kinetically stopped.

(c) It is helpful to report the number of atoms and the size of the modelled surfaces.

(d) In Fig. 6(c), how were the Fermi energies determined in the calculations? Does the calculated Fermi energy depend on the size of the supercell employed?

(e) Shown in Fig. 6(a), one O vacancy was introduced to the modelled CoO (111) surface. Have the authors considered different charge states of O vacancies? This could affect the stability and catalytic activity of the surface.

Therefore, I do not recommend this manuscript be published on Nature Communications as its current form.

Reviewer #3 (Remarks to the Author)

This manuscript demonstrated the surface atomic structure engineering of single-crystalline CoO nanorods by creating oxygen vacancies on their pyramidal nanofacets, which tailoring the electronic structure of CoO and resulting in excellent electrochemical activity. The work is quite interesting and the experimental and analysis results are informative and convincing. However, some requisite experiments and discussions need to be improved, such as the formation mechanism of surface O-vacancies. Accordingly, I'd like recommend the publication of this work after minor revisions, detailed suggestions are listed as following:

1. The precursive ZnO nanorods played an important role on the formation of SC CoO NRs, were there any same holes or nanopyramidal structures on the surface? Corresponding HRTEM images should be provided.

2. How did the O-vacancies arise during the cation exchange reaction? Please give further explanation.

3. The XANES spectra of SC CoO NRs have confirmed their rich O-vacancy surface, however, these O deficiency seemed indistinguishable in High-resolution HAADF-STEM image (as shown in Figure 2c), I suggest to highlight them by colored circles.

4. In figure 3b, the peak at ~ 800 eV of SC CoO NRs shift toward low photon energy, indicating the electron transfer from O-vacancies to Co. Did the Co 2p XPS spectra of SC CoO NRs agree with it?

5. Is the overall electrode activity of as-prepared SC CoO NRs comparable to the recently reported LDHs-based bifunctional materials (e.g. Adv. Energy Mater. 2015, 1500245)?

Response to Reviewer #1

General Comments:

In this manuscript, the authors report the surface atom structure tuning of single-crystal CoO nanorods through creating desired facets and oxygen defects, which show superior ORR and OER activities. Both experiments and calculations have been applied to explain the performances. This study is novel and very interesting. The data and methodology are reliable and the manuscript is also well organized and presented clearly. The conclusion is also clear and reliable. This manuscript can be considered for publication after addressing the following issues.

Response:

We would like to thank the Reviewer for his/her valuable comments and positive recommendation.

Original comment 1-1:

In Figure S9, the "PS CoO NRs" should be replaced by "SC CoO NRs".

Response:

We thank the Reviewer for pointing out this mistake. We have changed 'PS CoO NRs' to 'SC CoO NRs' in the original Fig. S9 (**Fig. S11** in the revised supporting information).

Original comment 1-2:

Figure 4c shows that SC CoO NRs have a better durability than Pt catalyst. However, as shown in Figure S11, after several hours testing the surface state has changed slightly. I want to know whether much O-vacancies can still be maintained in oxygen-saturated KOH. The O1s spectra of SC CoO NRs after several hours should also be supplied to compare the surface state.

Response:

The morphology change of SC CoO NRs is due to improper sample processing instead of instability of SC CoO NRs during ORR/OER reactions. Previously, after the durability test, SC CoO NRs were immediately taken for SEM observation. Because SC CoO NRs still adsorb large quantities of water and reactants, the surface of SC CoO NRs is easily oxidized and damaged by high energy electrons during analysis. To avoid such problem, the SC CoO NRs samples were sufficiently dried before SEM observations, and after drying the observed change in morphology is

much smaller (Fig. R1).

Figure R1. SEM images of SC NRs after 3000 CV cycle catalytic tests with accelerated scan rate of 100 mV/s in ORR region (0.5-1.1 V_{RHE}).

Moreover, according to the suggestion of the Reviewer, O1s spectra of SC CoO NRs before and after durability test were recorded. As shown in Fig. R2, the peak II ascribed to O-vacancies in SC CoO NRs after durability test is much higher than that before durability test, which is due to oxygen species adsorbed onto the surface of SC CoO NRs. In order to quantify the change of O-vacancies during durability test, we measured the active surface area of SC CoO NRs before and after durability test. As seen in Fig. R3, SC CoO NRs retained 95% of active surface area after durability test, which agrees very well with the durability test that SC CoO NRs maintained 97% of the reaction current after durability test (Fig. 4c). Since DFT calculations reveal that O-vacancies are active sites for ORR/OER (Fig. 6b), it is therefore concluded that the majority of O-vacancies are maintained during the ORR/OER.

Figure R2. XPS O 1s spectra of SC CoO NRs (a) before and (b) after stability test. The circles and lines denote experimental and deconvolution data, respectively.

Figure R3. Determination of active surface area of SC CoO NRs (a) before and (b) after durability test. (c) The ratio (R) of active surface area before and after durability test.

Following the Reviewer's comment, we have replaced original Fig. S11a (**Fig. S14a** in the revised supporting information) with Fig. R1 in the supporting information to avoid misleading and added the sample processing details in the **legend of Fig. S14**.

'Note that prior SEM analysis the samples of SC and PC CoO NRs were well dried in vacuum.'

Moreover, we have added Fig. R3 as **Fig. S13** in the revised supporting information.

Original comment 1-3:

The samples prepared by using 3h ZnO and 6h ZnO have the same length, but why SC CoO NRs using 3h ZnO exhibit the lowest onset potential and largest OER current as shown in Figure S16.

Response:

Although 3 h and 6 h ZnO NRs possess the same length, 3 h ZnO NRs have smaller average diameter with larger amount of holes across NRs (Fig. S5). This characteristic is highly beneficial for reactants diffusion inside NRs and guarantees a maximum exposure of the accessible active sites in the ORR/OER reactions. Therefore, SC CoO NRs exchanged with 3 h ZnO NRs as templates exhibit better performance.

Accordingly, we have added the related discussion in the legend of original Fig. S16 (**legend of Fig. S20** in the revised supporting information).

'A better performance of the 3 h sample in comparison to the 6 h sample (with nearly the same NR length) is mainly due to the presence of larger amount of holes in NRs of the 3 h sample (Fig. S5).'

Original comment 1-4:

In table S1, the atomic ratios of Co:O, Co:O_C and Co:O_L for SC and PC CoO NRs are quite different. The authors should add more discussions about them.

Response:

The quite different atomic ratios of Co:O, Co:O_C and Co:O_L for SC and PC CoO NRs originate from the uncommon O-vacancy rich nanofaceted surface structure of SC CoO NRs. The observed much smaller Co:O_C (1:2.19) ratio for SC CoO NRs as compared to that of PC CoO NRs (1:0.85) is ascribed to large quantity of water molecules and oxygen species adsorbed onto the {111}-Ov facets of SC CoO NRs, the surface energy of which is substantially higher than other low-indexed facets of CoO (*Nano Res.* 2010, 3, 1-7). The slightly smaller Co:O_L for SC CoO NRs is arising from the fact that 46% of the surface of SC CoO NRs is covered by pure O atoms terminated {111} facets. Please see the detailed analysis in **Note 2. XPS Analysis** in the supporting information.

Accordingly, to address this comment, we have added the related discussion in the revised **Note 2. XPS Analysis** in the supporting information.

‘Moreover, the much smaller Co:O_C ratio for SC CoO NRs as compared with that for PC CoO NRs (Supplementary Table S1) is ascribed to a large quantity of water molecules and oxygen species adsorbed onto the {111}-Ov facets of SC CoO NRs.’

Original comment 1-5:

*Some references about the effects of oxygen vacancy or crystal plane on electrocatalytic activity are missed. For examples:(1) *Angew. Chem. Int. Ed.* 2013, 52, 2474-2477. (2) *J. Mater. Chem. A* 2015, 3, 17598-17605. (3) *Nat. Chem.* 2012, 4, 1004-1010. (4) *ACS Catal.* 2016, 6, 400-406. (5) *Sci. Rep.* 2014, 4, 5767. (6) *J. Phys. Chem. C* 2015, 119, 4516.*

Response:

Following the Reviewer’s suggestion, we have included these six (6) references as **references [23]-[28]** in the revised manuscript.

Response to Reviewer #2

General Comments:

In the manuscript, the authors reported their experimental and computational results on electrocatalytic activity of single-crystal CoO nanorods terminated by O-vacancy-rich (111) facets for oxygen reduction reactions (ORR) and oxygen evolution reaction (OER) in alkaline solutions. Specifically, the authors had synthesized the CoO nanorods from ZnO nanorods using a cation exchange method, characterized the surface structure of the CoO nanorods using advanced electron microscopy and spectroscopy, and measured the catalytic activity of the fabricated CoO nanorods for ORR/OER using three electrode cell. Moreover, the authors performed density functional theory (DFT) calculations to predict the free energy evolution of ORR/OER on various CoO surfaces and the electronic structure of the CoO (111) surface with/without O vacancies. The authors concluded that the introduction of surface O vacancies could improve the electrocatalytic activity of CoO nanostructures. It is commendable that the authors have conducted this study in a meticulous way. However, the presented results only add some incremental knowledge/data to the current understanding of electrocatalysis. This reviewer does not believe the current manuscript contain enough innovative, significant contents to be considered for publication in Nature Communications.

Response:

We would like to thank the Reviewer for his/her valuable comments to help us to improve the quality of this manuscript.

Original comment 2-1:

It is unclear what the significances of this presented work are.

(1) As revealed in Fig. 4, the electrocatalytic activities of the synthesized CoO nanorods are worse than the state-of-the-art electrocatalyst systems. Especially, the authors studied the electrocatalytic properties of the CoO nanorods only in alkaline solutions, in which many other non-precious catalysts exhibit similar activities for ORR/OER.

Response:

We agree with the Reviewer that the ORR activity of as-synthesized SC CoO NRs currently do not exceed that of state-of-art Pt catalysts. However, their OER activity exceeds that of RuO₂ catalysts; their overall oxygen electrode activity is comparable with the well-developed cobalt oxide nanoparticles/N-doped graphene hybrid catalysts (*Nat. Mater.* 2011, 10, 780-786), which is considered as the most efficient bifunctional ORR/OER catalyst. The as-synthesized SC CoO NRs

have shown great potential in electrocatalysis. We believe that the catalytic activity of our SC CoO NRs can be further enhanced through increasing the mass loading (the current mass loading of SC CoO NRs on carbon fiber paper is only ~ 0.2 mg/cm², while that of most reported catalysts is more than 1 mg/cm²) and promoting the electronic conductivity via element doping.

Perhaps in the original submission we failed to underline the significance and novelty of this work. Therefore, we would like to highlight them below:

1. This work represents the first attempt to establish the atomic structure-function relationships for transition metal oxides (TMO). Based on the experimental studies and density functional theory (DFT) calculations, we demonstrate that the activity of CoO increases in the following order: {100}<{110}<{111}-Ov. There is a strong correlation between activity and atomic structure, the relationship that to our knowledge has not been considered in the analysis of the aqueous ORR/OER on any oxide surface. Given that the establishment of the atomic structure-function relationships for noble metals led directly to the development of the most efficient noble metal alloy electrocatalysts, the study of this crucial relationship for cost-effective TMOs is of broad scientific significance and great technological importance.

2. This work represents the first report on the TMO nanorods decorated with textured active nanofacets. Theoretical and experimental attempts have demonstrated that exposing more catalytically active facets is the most promising route to fully optimize the catalytic properties for noble metals and metal oxide catalysts. However, it is particularly challenging to expose these active facets due to their high surface energy. For CoO, the surface energy of {111} facets is much higher than that of {001} and {110} facets (*Nano Res.* 2010, 3, 1-7). Therefore, it is rather difficult to synthesize CoO exposed with {111} facets using traditional methods. Moreover, we demonstrate that oxygen vacancies are easy to be formed on {111} facets of CoO. Certainly, the as-synthesized SC CoO NRs enclosed with vacancy-rich {111} facets are highly preferential for eletrocatalytic applications.

Accordingly, to address this comment, we have added the related discussion in the revised manuscript on **page 9, lines 9-12,**

‘Our results demonstrate that there is a strong correlation between activity and atomic structure of CoO; that is, the ORR/OER activity of CoO increases in the following order {100}<{110}<{111}-Ov. To our knowledge this atomic scale structure-function relationship has not been considered for any

other TMO surfaces in the analysis of electrocatalysts.'

and on **page 5, lines 5-11**, to highlight the novelty of our work.

'Notably, for CoO the surface energy of {111} is much higher than that of other low-indexed facets⁴⁴. Such high percentage of {111} facets without foreign stabilizer is rather difficult to achieve via thermodynamic-controlled synthesis⁴⁴. However, in our kinetics-governed cation exchange strategy, facets with high surface energy and defects (discussed later) are forced to be exposed to facilitate the ion exchange process⁴⁵, assuring the formation of a large amount of clean and defect-rich {111} facets on the surface of SC CoO NRs, which is certainly highly preferable for catalysis.'

Moreover, we have added the above mentioned two references as **references [44] and [45]** in the revised manuscript.

Original comment 2-2:

(2) The employed experimental and computational techniques are widely used in the catalyst study.

Response:

Although the cation exchange method has been employed to fabricate nanomaterials, which have wide applications in photoluminance, solar cells, and so on, it has been seldom used to fabricate electrocatalysts. It should be noted that in traditional synthetic methods, foreign species, gas molecules, or surfactants are usually introduced to stabilize the active facets with high surface energy, which may affect the activity or the stability of the catalysts. In contrast, our synthetic route produces SC CoO NRs with high energetic and vacancy-rich {111} facets without any stabilizer. Certainly, these structural characteristics of the as-synthesized SC CoO NRs are highly attractive in electrocatalytic applications. Moreover, this methodology can be extended to synthesize NiO, FeO and so on, which have important applications in electrocatalysis. Therefore, we believe this report will attract a broad interest in the fields of nanotechnology and sustainable energy.

Regarding the computation part, it is true that we have applied a reliable computational technique to analyze the origin of the enhanced activity of SC CoO NRs toward ORR/OER, which can be easily reproduced by other researchers. Complementary with experimental observation, the DFT calculations reveal how reactants and intermediates interact with different facets of CoO in atomic level and identify the active sites of ORR/OER reactions on the surface of CoO. The significance and novelty of the DFT calculations lies in that:

1. This is the first study on the atomic level structure-function relationships on TMO in electrocatalysis. For details see our **response to comment 2-1**.

2. This study sheds some light on a new way of tuning O-vacancies on the specific facets.
The tuning of O-vacancies is of significant importance to modulate the electronic structure, thus to tailor the reactivity of metal oxides. Traditionally, annealing in reducing atmosphere and strong reducing treatment are commonly employed routes. In this study, we demonstrate that the high energetic {111}-O facets can stabilize large quantities of O-vacancies owing to their ultralow O-vacancy formation energy (Fig. 3c). Therefore, we propose a new route to tune the O-vacancies of metal oxides through exposing in a controlled way the crystal facets.

Following the Reviewer's comment, we have added the related discussion as listed in **our response to comment 2-1** in the revised manuscript to highlight the novelty of our work. *Moreover*, one sentence has been added in the revised manuscript on **page 6, lines 7-8**.

'It indicates that the surface defects can be tuned and stabilized through facet engineering.'

Original comment 2-3:

(3) The authors could not precisely control the concentration and distribution of the O vacancies on the surface of the CoO nanorods.

Response:

Actually, we have attempted to control the concentration of O-vacancies on the surface of SC CoO NRs through tuning the cation exchange temperature (Fig. R4). The structural and compositional information on the SC CoO NRs obtained via cation exchange at different temperatures is summarized in Table R1. As can be seen, an elevated temperature of the cation exchange process favors a high concentration of O-vacancies. Notably, the cation exchange temperatures lower than 600 °C cannot assure a complete conversion of ZnO into CoO. The ORR activity measurements show that SC CoO NRs obtained via cation exchange at 600 °C outperform the samples prepared at 650 and 700 °C with higher concentration of O-vacancies (Fig. R5a). HRTEM (Fig. R6) and electrochemical impedance spectroscopy (EIS, Fig. R5b) characterizations reveal that the high cation exchange temperature damages the surface and decreases the electronic conductivity of SC CoO NRs. Our results are in good agreement with previous reports that the formation of excessive oxygen defects would inevitably cause the structural instability and decrease

the electronic conductivity of metal oxides (*Angew. Chem. Int. Ed.* 2013, 52, 2474-2477; *Inorg. Chem.* 2014, 53, 9106-9114; *Adv. Mater.* 2015, 27, 5989-5994). These collective results demonstrate that the most efficient CoO NRs can be obtained via cation exchange at 600 °C with a modest concentration of O-vacancies.

Accordingly, to address this comment, we have added Fig. R4-R6, Table R1 as **Fig. S21-23**, **Table S7** and discussion on the control of O-vacancies in SC CoO NRs as **an note 5** in the revised supporting information: **Tuning the concentration of O-vacancies in SC CoO NRs**.

Figure R4. XRD spectra of SC CoO NRs obtained via cation exchange at different temperatures.

Table R1. Oxygen vacancy concentration, δ , of SC CoO NRs exchanged at different temperatures.

Sample	Cation exchange temperature (°C)	Lattice constant ^a a (Å)	Composition	δ ^b
1	600	4.234	CoO _{0.97}	0.03
2	650	4.236	CoO _{0.91}	0.09
3	700	4.245	CoO _{0.82}	0.18

^aFrom XRD measurement.

^bAverage data from inductively coupled plasma mass spectrometry (ICP-MS) measurement.

Figure R5. (a) ORR polarization curves and (b) EIS spectra (measured at 0.7 V_{RHE}) of SC CoO NRs obtained via cation exchange at different temperatures.

Figure R6. (a) and (b) Typical low magnification TEM and HRTEM images of CoO NRs obtained via cation exchange at higher temperatures, respectively.

Original comment 2-4:

(4) CoO is a semiconductor material which is not the best choice for electrocatalysts that requires rapid electron transfer.

Response:

We agree with the Reviewer that a rapid electron transfer is of crucial importance in electrocatalysis. Actually, the inherently poor electronic conductivity of metal oxides is one of the key problems that hampers their activity and stability (*Nat. Commun.* 2014, 5, 4191), not only in relation to CoO. We found that the large quantities of O-vacancies localized on the {111} facets as well as the single-crystal structure inherited from the ZnO NRs can significantly improve the

intrinsic electronic conductivity of the as-synthesized SC CoO NRs. As shown in Fig. R7, the carrier concentration in SC CoO NRs is one order higher than that of poly-crystalline (PC) CoO NRs. We believe that the conductivity of SC CoO NRs can be further enhanced through element doping.

Figure R7. Mott-Schottky (M-S) plots of SC and PC CoO NRs.

On the other hand, it was shown recently that CoO is highly active for either ORR (*J. Am. Chem. Soc.* 2012, 134, 15849-15857; *Nat. Commun.* 2013, 4, 1805) or OER (*Nat. Commun.* 2015, 6, 7261). Our primary evaluation of the activity and durability of SC CoO NRs clearly demonstrates that the as-synthesized SC CoO NRs are highly versatile and efficient electrocatalysts toward ORR/OER. Moreover, our DFT calculations reveal that the {111}-Ov facets of CoO are highly active toward ORR/OER. Therefore, CoO and Co-based oxides are promising materials for electrocatalysis.

Following the Reviewer's comment, we have added Fig. R7 as **Fig. S15** in the revised supporting information. *Moreover*, we have added a new section to discuss the electronic conductivity enhancement of SC CoO NRs in the revised manuscript **on page 7, lines 11-21: *Enhancement of electronic conductivity of SC CoO NRs. High activity and durability of SC CoO NRs clearly demonstrates that the as-synthesized SC CoO NRs are highly versatile and efficient electrocatalysts toward both ORR and OER. One prerequisite in the design of a highly efficient electrocatalyst is a rapid electron transfer¹⁶. It is acknowledged that the electronic conductivity of TMOs is relatively poor¹⁵, greatly limiting their electrocatalytic activities. In the case of SC CoO NRs, a large quantity of O-vacancies localized on the {111}-O facets as well as the single-crystal***

nature inherited from the ZnO NRs considerably enhance the carrier concentration in SC CoO NRs, which is one order higher than that in PC CoO NRs (Supplementary Fig. S15 and Note 3). Moreover, the nucleation and growth of SC CoO NRs directly on CFP also assures a rapid collection of charges. The aforementioned three important characteristics remarkably enhance the electronic conductivity of SC CoO NRs.’

Moreover, **Note 3. Mott-Schottky Analysis** has been added in the revised supporting information to describe the analysis of the M-S plots to quantify the carrier concentration in the as-synthesized SC CoO NRs.

‘Note 3. Mott-Schottky Analysis

The acceptor density can be calculated from the slopes of the M-S plots (Supplementary Fig. S15) by the following equation³¹,

$$\frac{dC^{-2}}{dV} = \frac{-2}{q\epsilon_0\epsilon_r N_A A^2} \quad (S13)$$

where A is the surface area of the sample studied, ϵ_r is the dielectric constant of CoO with the value of 5.4. The acceptor concentrations in SC and PC CoO NRs were estimated to be 3.8×10^{20} and $2.4 \times 10^{19} \text{ cm}^{-3}$, respectively, indicating a greatly enhanced carrier concentration in SC CoO NRs provided by the single-crystalline nature and the O-vacancies present on the surface of SC CoO NRs.’

Original comment 2-5:

It is not a novel idea to tune the properties of semiconductor nanostructures through modification of their surface defects. Specifically for electrocatalysts, there are two main mechanisms for performance improvement. Surface defects might (a) enhance the electrical conductivity and (b) alter the paths of chemical reactions. The authors mentioned both effects in the manuscript but did not carry out sufficient study to quantify the two effects and pin down the main reason for the observed electrocatalytic activity enhancement. Therefore, this reviewer does not think the current manuscript is novel enough to be of interests to a broad audience of scientific community.

Response:

In principle, both (a) and (b) effects are related to the changes in electronic structure of the material, which rise up simultaneously as the result of generation of O-vacancies on the {111}

facets of SC CoO NRs. Since these effects have the same physical origin, it is hard to separate and quantify them. We think that these two effects contribute significantly to the observed enhancement of electrocatalytic activity, which has been experimentally confirmed. As shown in Fig. 6b, the creation of O-vacancies on the {111} facets enables the optimal adsorption energies for ORR/OER intermediates, which directly results in smaller Tafel slopes toward ORR/OER (Fig. 4, Fig. S16 and Fig. S17). Moreover, as discussed in **our response to comment 2-4** and shown in Fig. R7, O-vacancies are indeed helpful to considerably increase the carrier density, thus to enhance the electron transfer in the as-synthesized SC CoO NRs (Fig. S19). Therefore, (a) and (b) together promote the reactivity of SC CoO NRs. Otherwise, this metal oxide cannot achieve such excellent performance as other well-developed catalysts (Table S2).

The novelty of this work has been discussed in details in **our response to comments 2-1~2-4**.

Following the Reviewer's comment, we have added in the revised manuscript the related discussion as listed in **our response to comments 2-1~2-4** to highlight the novelty of our work.

Original comment 2-6:

This manuscript does not contain durability test results for the CoO nanorods. It happens frequently that so-called "surface engineered" nanocatalysts exhibit initial good activity but lose their activity rapidly in electrochemical cycling tests.

Response:

We have shown the durability of the as-synthesized SC CoO NRs in Fig. 4c, which confirms much better durability of SC CoO NRs as compared to the state-of-art Pt catalysts. In addition, we measured the active surface area of SC CoO NRs before and after durability test. As shown in Fig. R8, after 10 h durability test, SC CoO NRs still retained 95% of the active surface area, which agrees very well with the durability test indicating that SC CoO NRs retained 97% of the initial reaction current (Fig. 4c). As discussed in the manuscript, this excellent durability originates from direct growth of SC CoO NRs on the CFP substrate to avoid aggregating and detaching problems, which are usually encountered in the case of other 'surface engineered' nanocatalysts as mentioned by the Reviewer. The long-term stability of the as-synthesized SC CoO NRs highlights their great potential in electrocatalytic applications.

Figure R8. Determination of active surface area of SC CoO NRs (a) before and (b) after durability test. (c) The ratio (R) of active surface area before and after durability test.

Accordingly, to address this comment, we have added Fig. R8 as **Fig. S13** in the revised supporting information.

Original comment 2-7:

Regarding the DFT computation, I have several questions about some deficiencies related to the computational details presented in the manuscript and Supplementary Information.

(a) It has been reported that spin-polarization is needed for surface adsorption calculation. The authors did not mention if they include this in their DFT calculations.

Response:

All results are based on the spin-polarized calculations. We have added the related description in the **Computation Section** of the revised supporting information.

*‘All structures in the calculations were **spin-polarized** and relaxed until the convergence tolerance of force on each atom was smaller than 0.05 eV.’*

Original comment 2-8:

(b) It is unclear how the authors compared the predicted electrocatalytic activities for ORR/OER on different CoO surfaces from Fig. 6(b). At the given potential 1.23 V, all the electrochemical reactions should have kinetically stopped.

Response:

The comparison of the predicted electrocatalytic activity is based on the following equation

proposed by Nørskov *et al.* (*J. Phys. Chem. B* 2004, 108, 17886-17892),

$$i_{k,0} = \tilde{i}_k \exp[-\Delta G(U_0) / kT] \quad (\text{R1})$$

where $i_{k,0}$ is the exchange current and $\Delta G(U_0)$ is the maximum free energy change as shown on the free energy diagram. Briefly speaking, the lower free energy change on the diagram, the faster the reaction kinetics (at any electrode potential). As discussed in the manuscript, the ideal reaction pathways for ORR/OER should possess the minimum overall reaction free energy change ($|\Delta G_{\text{OOH}^*}| \approx |\Delta G_{\text{O}^*}| \approx |\Delta G_{\text{OH}^*}| \rightarrow 0$) at the thermodynamic equilibrium potential (*Chem. Soc. Rev.* 2015, 44, 2060). Since {111}-Ov of CoO exhibits the lowest overall free energy change, it exhibits highest reaction kinetics and therefore the highest exchange current density, one of the most important activity descriptors.

It is true that at the equilibrium potential, 1.23 V_{RHE}, the overall electrochemical reaction as represented by current density is zero, which appears to be ‘stopped’. In fact this overall current density is the sum of the ‘forward reaction (i.e. ORR)’ and ‘backward reaction (i.e. OER)’; both of these components are not zero and are represented by the corresponding exchange current density (*Chem. Soc. Rev.* 2015, 44, 2060-2086).

Original comment 2-9:

(c) *It is helpful to report the number of atoms and the size of the modeled surfaces.*

Response:

All periodic slabs have a vacuum spacing of at least 15 Å. The structural model of CoO {100} facet contains four Co-O layers (128 atoms), with a supercell size of $a=b=12.07\text{Å}$, $c=21.80\text{Å}$, $\alpha=\beta=\gamma=90^\circ$, while {110} facet consists of four Co-O layers (64 atoms), with $a=8.53\text{Å}$, $b=12.07\text{Å}$, $c=20.05\text{Å}$, $\alpha=\beta=\gamma=90^\circ$. The {111}-O surface model is composed of four Co layers and five O layers (160 atoms), with $a=b=12.07\text{Å}$, $c=29.78\text{Å}$, $\alpha=\beta=90^\circ$, $\gamma=120^\circ$, and the {111}-Ov surface is obtained by removing one surface O atom (159 atoms). For {111}-O and {111}-Ov surfaces, one H layer was introduced to make the slab to obey the electron counting rule, which is a common method used for polar surfaces (*J. Phys.: Condens. Matter* 2014, 26, 315014; *J. Chem. Phys.* 2013, 139, 124704). The top-view and side-view of the models are shown in Fig 6a. In calculations, the two bottom

layers (plus hydrogen atom layer for {111}-O and {111}-Ov) were kept fixed, whereas the rest of atoms were allowed to relax.

Accordingly, we have added the above description of surface models in the **Computation Section of Supplementary Methods** in the revised supporting information.

Original comment 2-10:

(d) In Fig. 6(c), how were the Fermi energies determined in the calculations? Does the calculated Fermi energy depend on the size of the supercell employed?

Response:

The Fermi levels are determined by using the following equation:

$$N = \int_{-\infty}^{E_f} O(e) de \quad (\text{R2})$$

where the integration of the occupation numbers ($O(\epsilon)$) from negative infinity to Fermi energy (E_f) gives the total number of electrons (N) in the system. For example, see the valance band maximum in the upper panel of Fig. 6c and the energy of band gap state in the lower panel of Fig. 6c. The calculated Fermi energy doesn't depend on the size of the supercell employed for system without vacancies, which obeys the fundamental rule of periodic systems. However, in the system with vacancies, it may vary when size of the supercell changes, since the concentration of the vacancies changes.

Original comment 2-11:

(e) Shown in Fig. 6(a), one O vacancy was introduced to the modeled CoO (111) surface. Have the authors considered different charge states of O vacancies? This could affect the stability and catalytic activity of the surface.

Response:

We did not consider the different charge states of O-vacancies in the calculation of free energy diagram. To the best of our knowledge, the calculation for a system with charged defects is only suitable for bulk materials (*J. Appl. Phys.* 2004, 95, 3851). More specifically, in the VASP manual (section 6.64 Monopole, Dipole and Quadrupole corrections), there is some text relevant for calculations of a charge slab (surface model): "It is important to emphasize that the total energy cannot be corrected for charged slabs, since a charged slab results in an electrostatic potential that

grows linearly with the distance from the slab (corresponding to a fixed electrostatic field). It is fairly simple to show that as a result of the interaction between the charged slab and the compensating background, the total energy depends linearly on the width of the vacuum. Presently, no simple a posteriori correction scheme is known or implemented in VASP. **Total energies from charged slab calculations are hence useless, and cannot be used to determine relative energies.**” Therefore, we didn’t provide those data, which can be misleading.

In the calculation of the free energy diagram (Fig. 6b), we included the effect of a bias involving in the electrode, by shifting the free energy by $-eU$, where U is the electrode potential (*J. Phys. Chem. B* 2004, 108, 17886-17892; *J. Am. Chem. Soc.* 2014, 136, 4394-4403). The different charge states of O-vacancies should be merged in the potential U in the calculations.

As regards to the stability of {111}-Ov facets, the durability test (Fig. 4c) and the measurement of active surface area (Fig. R8) confirm that these facets are stable during catalytic reactions, which is more convincing than the results from calculations.

Response to Reviewer #3

General Comments:

This manuscript demonstrated the surface atomic structure engineering of single-crystalline CoO nanorods by creating oxygen vacancies on their pyramidal nanofacets, which tailoring the electronic structure of CoO and resulting in excellent electrochemical activity. The work is quite interesting and the experimental and analysis results are informative and convincing. However, some requisite experiments and discussions need to be improved, such as the formation mechanism of surface O-vacancies. Accordingly, I'd like recommend the publication of this work after minor revisions, detailed suggestions are listed as following:

Response:

We would like to thank the Reviewer for his/her helpful comments and positive recommendation.

Original comment 3-1:

The precursive ZnO nanorods played an important role on the formation of SC CoO NRs, were there any same holes or nano-pyramidal structures on the surface? Corresponding HRTEM images

should be provided.

Response:

As shown in Fig. R9, ZnO NRs exhibit regular hexagonal shapes without any holes inside NRs, and the side surface of ZnO NRs is enclosed with smooth $\{10\bar{1}0\}$ planes. We found that the cation exchange (replacement of Zn^{2+} with Co^{2+}) is rather a kinetically controlled process not thermodynamically controlled one. Hence, high energy $\{111\}$ facets of CoO were forced to be exposed to facilitate the cation exchange process. To balance the total surface energy, $\{100\}$ facets with relatively low surface energy are exposed too. Therefore, the aforementioned cation exchange methodology produces CoO NRs enclosed with textured nanopyramids.

Figure R9. (a) Top-view SEM image of ZnO NRs. (b) HRTEM image of a single ZnO NR with the low magnification TEM image displayed in the inset, showing the smooth surface of initial ZnO template.

Following the Reviewer's comment, we have added Fig. R9 as **Fig. S2** in the revised supporting information.

Original comment 3-2:

How did the O-vacancies arise during the cation exchange reaction? Please give further explanation.

Response:

The creation of vacancies during ion exchange process has been well investigated, for instance,

the exchange of Pb for Cd on the surface of PbS (*ACS Nano* 2014, 8, 7948-7957). As shown in Fig. R10, the empty S surface sites facilitate the exchange of Pb for Cd, resulting in the formation of an initial CdS shell. The existing vacancies control the exchange process (*Chem. Soc. Rev.* 2013, 42, 89-96). A similar situation is in the system studied, where a large quantity of O-vacancies present on the surface of SC CoO NRs facilitates the exchange of Zn for Co.

Figure R10. Schematic representation of the exchange of Pb for Cd on the surface of PbS quantum dots (*ACS Nano* 2014, 8, 7948-7957).

Following the Reviewer’s comments 3-1 and 3-2, we have added the related discussion about the formation of pyramidal nanofacets and O-vacancies on the surface of SC CoO NRs in the revised manuscript **on page 5, lines 5-11**.

‘Notably, for CoO the surface energy of {111} is much higher than that of other low-indexed facets⁴⁴. Such high percentage of {111} facets without foreign stabilizer is rather difficult to achieve via thermodynamic-controlled synthesis⁴⁴. However, in our kinetics-governed cation exchange strategy, facets with high surface energy and defects (discussed later) are forced to be exposed to facilitate the ion exchange process⁴⁵, assuring the formation of a large amount of clean and defect-rich {111} facets on the surface of SC CoO NRs, which is certainly highly preferable for catalysis.’

Moreover, we have added two aforementioned references as **[44] and [45]** in the revised manuscript.

Original comment 3-3:

The XANES spectra of SC CoO NRs have confirmed their rich O-vacancy surface, however, these O deficiency seemed indistinguishable in High-resolution HAADF-STEM image (as shown in Figure

2c), I suggest to highlight them by colored circles.

Response:

In our HAADF-STEM experiment, the convergent semiangle and collection angle were chosen as 21.5 and 200 mrad, respectively. Under this condition, the high angle annular detector (Fig. R11) cannot detect electrons scattered from both Co and O atoms due to relative low electron scattering capability of O atoms. Therefore, the bright contrasts in Fig. 2c and 2e are only Co atomic columns, and O atoms cannot be distinguished in the HAADF-STEM image.

Figure R11. Schematic diagram of HAADF-STEM.

Original comment 4:

In figure 3b, the peak at ~800 eV of SC CoO NRs shift toward low photon energy, indicating the electron transfer from O-vacancies to Co. Did the Co 2p XPS spectra of SC CoO NRs agree with it?

Response:

The Co 2p XPS spectra of the as-synthesized SC CoO NRs and commercial CoO powder (reference CoO) agree well with the XANES results. As shown in Fig. R12, a slight shift of Co 2p satellite peak towards low binding energy direction (indicated by the arrow), reveals the reduced Co²⁺ state on the surface of SC CoO NRs due to the electron transfer from O-vacancies.

Figure R12. Co 2p XPS spectra of SC CoO NRs and commercial CoO powder (reference CoO).

Following the Reviewer’s comment, we have added Fig. R12 as **Fig. S8** in the revised supporting information and related discussion in the revised manuscript **on page 5, lines 21-24**.

‘This is also consistent with the observation of a noticeable peak shift in Co-L_{2,3} edge toward low photon energy and Co 2p XPS spectrum toward low binding energy of SC CoO NRs (Fig. 3b and Supplementary Fig. S8), which is an indicative of electron transfer from O-vacancies to Co d band.’

Original comment 3-5:

Is the overall electrode activity of as-prepared SC CoO NRs comparable to the recently reported LDHs-based bifunctional materials (e.g. Adv. Energy Mater. 2015, 1500245)?

Response:

Following the Reviewer’s suggestion, we have compared the overall electrode activity of the as-synthesized SC CoO NRs with LDHs-based bifunctional catalyst in **Table S2** in the revised supporting information and added the reference mentioned by the Reviewer as **reference [19]** in the revised manuscript to confirm the excellent activity of the as-synthesized SC CoO NRs as compared with other well-developed bifunctional catalysts.

END OF RESPONSE

Reviewers' Comments:

Reviewer #1 (Remarks to the Author)

All the concerns have been well addressed and the manuscript has also been improved. I think the present manuscript can be accepted for publication.

Reviewer #2 (Remarks to the Author)

The manuscript has been greatly improved. All my previous concerns have been properly addressed. Therefore, I recommend this manuscript for publication in Nature Communications in its current form.

Reviewer #3 (Remarks to the Author)

The manuscript has been revised according to reviewers' suggestion. Author's response is satisfactory. I would recommend the publication of this work.

Response to the Editor

EDITORIAL REQUESTS:

1. * *Due to our policy on transparent peer review, please clearly state in your cover letter whether you wish to opt in or out for the publication of the reviewer reports.*

Response:

We agree to publish the reviewer comments and our corresponding response online as a supplementary peer review file and clearly state in our cover letter.

2. * *Please note that the Multimedia License to Publish form only applies to the thumbnail or featured image. Please use one form for each item and change the title on the form to a short scientific description of the chosen image. Please do not use the manuscript title, 'cover art', 'thumbnail' or 'featured Image'.*

Response:

Done.

3. * *Sorry, it appears that we sent you the wrong payment form - please find the correct one attached.*

Response:

Done.

4. * *Please number the Supplementary Notes 'Supplementary Note 1', 'Supplementary Note 2' etc in the Supplementary Information.*

Response:

We have numbered the supplementary notes in the revised Supplementary Information.

5. * *Please note that the Supplementary Methods should not be numbered but instead one text.*

Response:

We have corrected the related part in supplementary Methods in the revised Supplementary Information.

6. * *Please cite your Supplementary Fig 20 in the article.*

Response:

We have cited Supplementary Fig. 20 in **lines 9-10, page 10** in the Method Section.

“Synthesis of SC CoO NRs on CFP substrate. ZnO NRs with tailorable length (**Supplementary Figs. 4 and 20**) were grown on CFP under hydrothermal conditions and finally converted into CoO NRs using a cation exchange process in gas phase (Supplementary Fig. S1).”